# Impact of different visceral metastatic sites on survival in metastatic prostate cancer patients

**Gu-Shun Lai**[1,2], **Chuan-Shu Chen**[1,2,3], **Jason Chia-Hsien Cheng**[4], **Jian-Ri Li**[1,2,3,4,5], **Cheng-Kuang Yang**[1,2,6], **Chia-Yen Lin**[1,2,7], **Sheng-Chun Hung**[1,2,3], **Kun-Yuan Chiu**[1,2,8], **Shian-Shiang Wang**[1,2,8]*

1 Institute of Medicine, Chung Shan Medical University, Taichung, Taiwan, 2 Department of Urology, Taichung Veterans General Hospital, Taichung, Taiwan, 3 Department of Post-Baccalaureate Medicine, College of Medicine, National Chung Hsing University, Taichung, Taiwan, 4 Division of Radiation Oncology, Department of Oncology, National Taiwan University Hospital, Taipei, Taiwan, 5 Department of Medicine and Nursing, Hungkuang University, Taichung, Taiwan, 6 Jenteh Junior College of Medicine, Nursing and Management, Miaoli, Taiwan, 7 College of Medicine, National Yang Ming Chiao Tung University, Taipei, Taiwan, 8 Department of Applied Chemistry, National Chi Nan University, Nantou, Taiwan

* urologyssw@gmail.com

## Abstract

### Introduction

Visceral metastasis is an important predictor for poor outcomes in prostate cancer, however, the prognostic significance surrounding the specific sites of visceral metastasis remains unclear. The aim of this study was to evaluate the impact of different visceral metastatic sites on survival in patients with prostate cancer.

### Methods

We identified patients with metastatic prostate cancer between January 1, 2010 and December 31, 2023 using the TriNetX database. Patients were divided into 4 cohorts according to their specific metastatic sites: lung metastases, brain metastases, liver metastases, and bone metastases. Survival analysis was calculated using the Kaplan-Meier method and Cox regression models.

### Results

In total, 59,875 patients diagnosed with metastatic prostate cancer were identified, with 39,495 (65.2%) having bone metastases, 7,573 (12.5%) lung metastases, 5,240 (8.7%) brain metastases, and 7,567 (12.5%) liver metastases. The median overall survival was 44.4 months for patients with bone metastases, 31.9 months for lung metastases, 9.6 months for brain metastases, and 10 months for liver metastases. Lung metastases were associated with an improved survival when compared with liver and brain metastases. For patients with two visceral metastatic sites or concomitant bone metastases, liver metastases were related to worse outcomes. Asian patients experienced better OS than Caucasian and African American patients in visceral metastatic prostate cancer.

**Funding:** The author(s) received no specific funding for this work.

**Competing interests:** The authors have declared that no competing interests exist.

## Conclusion

Patients with lung metastases experienced better survival outcomes in prostate cancer with only one visceral metastatic site. Liver metastases were associated with worse outcomes when there were two visceral metastatic sites combined or concomitant bone metastases. Asian patients displayed improved survival rates when compared with both Caucasian and African American patients in visceral metastatic prostate cancer.

## Introduction

Prostate cancer is the fourth most frequently diagnosed cancer in men and the eighth leading cause of cancer death worldwide. More than 1.4 million prostate cancer cases were diagnosed, with 397,430 patients dying of the disease in 2022 [1]. Although there are good survival outcomes for those with localized disease, the risk of death still remains high for metastatic prostate cancer (mPC) patients. Several studies have demonstrated the adverse impact of visceral metastases on overall survival (OS) for patients with metastatic prostate cancer (mPC) [2–13]. However, information regarding the significance of specific visceral metastases on patient prognosis remains scarce. A meta-analysis taken from randomized clinical trials conducted by Halabi et al. identified lung and liver metastases, when compared with bone and other non-visceral metastases, as being risk factors for poor prognosis in patient diagnosed with metastatic castration resistant prostate cancer (mCRPC) [9]. Tappero et al., in a population-based analysis, reported on there being better outcomes for patients with lung metastases when compared with other visceral metastases [13]. Despite the results of these studies, there remains limited data regarding the prognostic significance of specific metastatic sites for patient diagnosed with metastatic prostate cancer.

Herein, we performed a retrospective analysis using the TriNetX database in order to investigate the influence of specific sites of visceral metastases on overall survival in metastatic prostate cancer patients. We also evaluated the outcomes of visceral metastases with regard to different non-visceral metastases, as well as to race.

## Methods

### Data source, study population, and outcomes

We utilized the TriNetX network to conduct a retrospective analysis. Data were retrieved from the US Collaborative Network, which includes 57 healthcare organizations across the United States. Data collection and analysis for this research were performed on the TriNetX platform in February, 2024.

Through the network we identified patients aged ≥18 years who had been diagnosed with mPC during the period of January 1, 2010 and December 31, 2023. The diagnosis of mPC was performed according to the International Classification of Diseases, tenth edition, Clinical Modification (ICD-10-CM): ICD-10-CM C61, as well as ICD-10-CM: C77, C78, C78.7, C79.3, or C79.5 in order to confirm the distant metastases. The index date was set to be the date of diagnosis of distant metastasis. Patients enrolled were divided into 3 cohorts according to the sites of visceral metastases: brain, liver, and lung metastasis, while patients with bone metastasis without visceral metastases were included for comparison. Enrolled patients could either have or not have non-visceral metastasis. The primary end point was OS, which was calculated from the date of diagnosis of distant metastases (the index date) to the date of death or

censored at the end of the study, whichever occurred first. We also performed survival analyses for patients with mPC among different races and concomitant non-visceral metastases.

### Statistical analyses

For analyzing patients baseline characteristics, mean with standard deviation (SD) was utilized for continuous variables, and number with percentage for categorical variables. Kaplan-Meier survival analysis along with the log-rank test and Cox proportional regression model were used for the evaluation of OS among the different cohorts. All statistical analyses were performed on the TriNetX network with a $p$ value <0.05 being considered statistically significant.

### Ethics in research

This research was carried out after receiving approval from the Institutional Review Board (IRB) of Taichung Veterans General Hospital (IRB number: SE:22220A). Given the TriNetX platform provides only de-identified data, the ethics committee approved a waiver of informed consent.

## Results

### Demographics and characteristics

A total of 59,875 patients with metastatic prostate cancer were identified on the TriNetX database. Of them, 39,495 (65.2%) had bone metastases, 7,573 (12.5%) lung only visceral metastases, 5,240 (8.7%) brain only visceral metastases, and 7,567 (12.5%) liver only visceral metastases. Compared to the patients in the bone metastases cohort, patients in the visceral metastases cohort were at a significantly younger age, possessed a poor Eastern Cooperative Oncology Group (ECOG) performance status, and had more lymph node metastases, comorbidities, and a higher proportion of receiving novel hormone agents or chemotherapy. Patient demographics and characteristics are demonstrated in Table 1.

**Table 1. Baseline characteristics for patients with visceral and bone metastatic prostate cancer.**

|  | Bone n = 39,495 | Lung n = 7573 | Brain n = 5240 | Liver n = 7567 |
|---|---|---|---|---|
| Age at index, years, mean (SD) | 73.7 (9.78) | 72.6 (10.4)* | 70.4 (9.95)* | 71.8 (9.71)* |
| Race, n (%) |  |  |  |  |
| Caucasians | 28,491 (71) | 5452 (72)* | 3825 (73)* | 5523 (73)* |
| African Americans | 5793 (15) | 985 (13)* | 681 (13)* | 1059 (14)* |
| Asians | 1028 (3) | 227 (3) | 157 (3) | 151 (2) |
| Others/unknown | 4183 (10) | 909 (12) | 577 (11) | 834 (11) |
| BMI at index, mean (SD) | 27.2 (5.53) | 26.8 (5.64)* | 26.2 (5.39)* | 26.4 (5.44)* |
| ECOG at index, mean (SD) | 0.924 (0.965) | 1.2 (0.942)* | 1.38 (1.01)* | 1.23 (1.04) |
| PSA at index, ng/ml, mean (SD) | 214 (1041) | 184 (663)* | 332 (1020)* | 288 (866)* |
| Non-visceral metastatic sites, n (%) |  |  |  |  |
| Lymph Node | 56 (3.5) | 1673 (19)* | 1263 (21)* | 1354 (16)* |
| Bone | 39,495 (100) | 2338 (26) | 2460 (40) | 2138 (26) |
| Systemic therapies, n (%) |  |  |  |  |
| Abiraterone | 2315 (6) | 660 (9)* | 758 (14)* | 853 (11)* |
| Enzalutamide | 1694 (7) | 509 (6.7)* | 568 (11)* | 671 (9)* |
| Apalutamide | 290 (0.7) | 67 (0.8) | 70 (1.3)* | 75 (0.9) |
| Docetaxel | 648 (1.6) | 446 (6)* | 628 (12)* | 662 (9)* |

(*Continued*)

**Table 1.** (Continued)

|  | Bone n = 39,495 | Lung n = 7573 | Brain n = 5240 | Liver n = 7567 |
|---|---|---|---|---|
| Cabazitaxel | 96 (0.2) | 98 (1)* | 198 (4)* | 209 (3)* |
| Comorbidity, n (%) |  |  |  |  |
| Diabetes Mellitus | 6923 (18) | 2383 (27)* | 1416 (23)* | 2348 (28)* |
| Hypertension | 16,748 (42) | 2390 (61)* | 3457 (57)* | 4915 (59)* |
| Cerebrovascular disease | 4044(10) | 1400 (16)* | 1153 (19)* | 1268 (15)* |
| Ischemia heart disease | 8086 (20) | 2920 (33)* | 1767 (29)* | 2498 (30)* |

BMI: Body Mass Index; ECOG, Eastern Cooperative Oncology Group; PSA, Prostate-Specific Antigen; SD, Standard Deviation.

* Statistical difference from the bone metastases group.

## Outcomes of overall population and different visceral metastatic sites

The median OS for mPC patients with visceral metastases was 15.7 months [95% confidence interval (CI) 15–16.2] and the survival probability at the 12$^{th}$ month was 24.7% (95% CI 54.1–55.5).

The median OS was 44.4 months (95% CI 43.1–45.5) for patients with bone metastases, 31.9 months (95% CI 29.8–34.4) for lung metastases, 9.6 months (95% CI 8.9–10.4) for brain

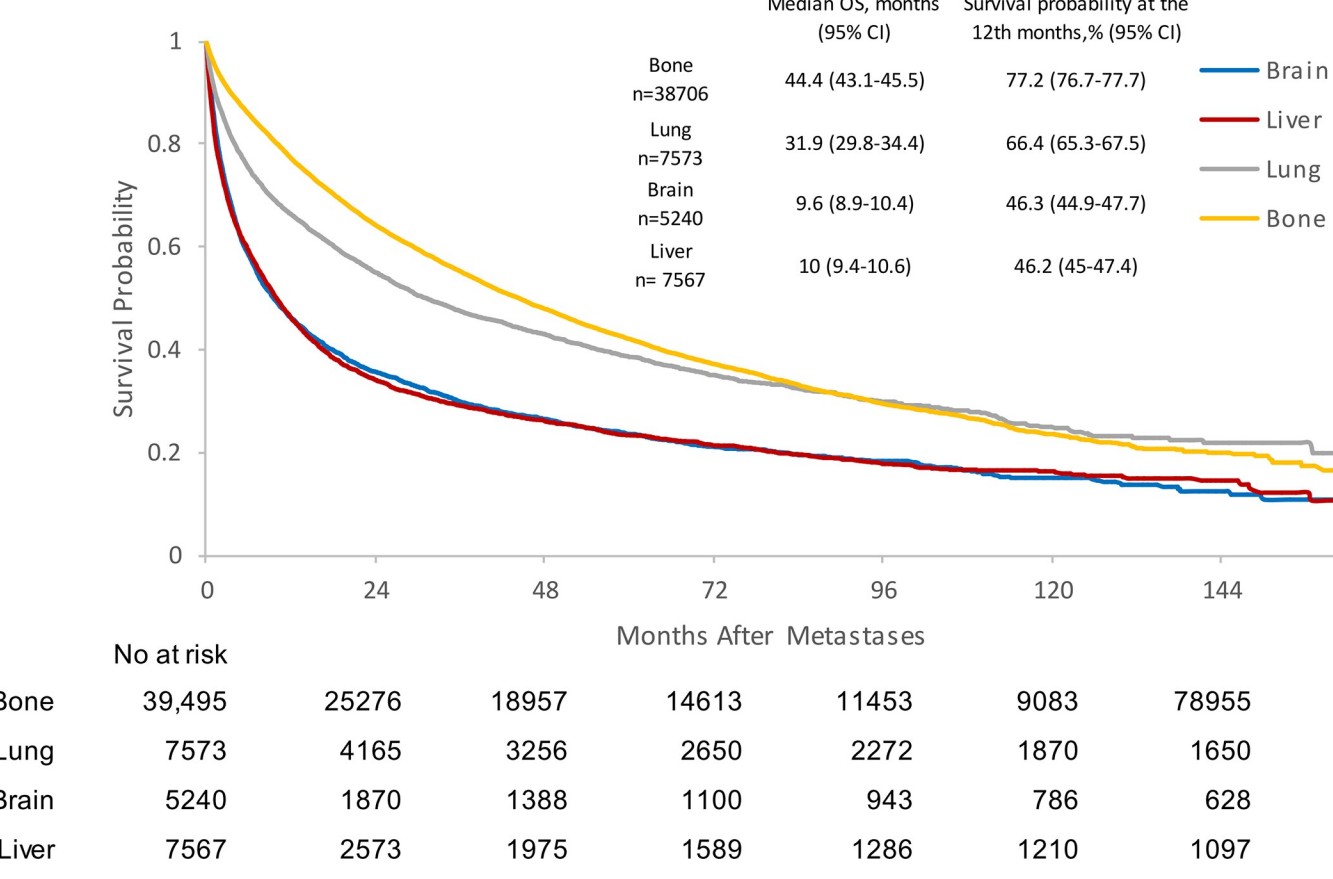

**Fig 1. Kaplan-Meier analysis of overall survival for prostate cancer patients with bone or one visceral metastatic site.** CI: confidence interval; OS: overall survival.

**Table 2. Cox regression analysis of overall survival for patients with visceral metastases.**

| | HR | 95% CI | | p-value | HR | 95% CI | | p-value |
|---|---|---|---|---|---|---|---|---|
| Site of metastases (one site) | | | | | | | | |
| Bone (n = 39,495) | Reference | | | | 0.775 | 0.748 | 0.802 | <0.0001 |
| Lung (n = 7,573) | 1.29 | 1.245 | 1.355 | <0.0001 | Reference | | | |
| Brain (n = 5,240) | 2.246 | 2.166 | 2.333 | <0.0001 | 1.671 | 1.595 | 1.749 | <0.0001 |
| Liver (n = 7,567) | 2.3 | 2.229 | 2.379 | <0.0001 | 1.692 | 1.623 | 1.765 | <0.0001 |
| Site of metastases (two sites) | | | | | | | | |
| Brain + Lung (n = 1,453) | Reference | | | | 0.878 | 0.813 | 9494 | 0.001 |
| Liver + Lung (n = 3,471) | 1.138 | 1.054 | 1.23 | 0.001 | Reference | | | |
| Brain + Liver (n = 716) | 1.707 | 1.529 | 1.906 | <0.0001 | 1.488 | 1.349 | 1.64 | <0.0001 |
| Visceral metastases with concomitant bone metastases | | | | | | | | |
| Lung + Bone (n = 2,604) | Reference | | | | 0.76 | 0.96 | 0.824 | <0.0001 |
| Brain + Bone (n = 1,620) | 1.319 | 1.213 | 1.436 | <0.0001 | Reference | | | |
| Liver + Bone (n = 2,418) | 1.643 | 1.524 | 1.772 | <0.0001 | 1.27 | 1.164 | 1.38 | <0.0001 |
| Visceral metastases with concomitant lymph node metastases | | | | | | | | |
| Lung + Lymph node (n = 1,307) | Reference | | | | 0.613 | 0.512 | 0.735 | <0.0001 |
| Brain + Lymph node (n = 282) | 1.63 | 1.361 | 1.952 | <0.0001 | Reference | | | |
| Liver + Lymph node (n = 998) | 1.706 | 1.513 | 1.922 | <0.0001 | 1.06 | 0.886 | 1.269 | 0.5205 |
| Visceral metastases with concomitant bone and lymph nodes metastases | | | | | | | | |
| Lung + Bone + Lymph node (n = 1,841) | Reference | | | | 0.723 | 0.643 | 0.812 | <0.0001 |
| Brain + Bone + Lymph node (n = 243) | 1.396 | 1.242 | 1.568 | <0.0001 | Reference | | | |
| Liver + Bone + Lymph node (n = 1,453) | 1.641 | 1.499 | 1.796 | <0.0001 | 1.187 | 1.056 | 1.334 | 0.004 |
| Race | | | | | | | | |
| Caucasians (n = 19,490) | Reference | | | | 1.081 | 1.031 | 1.135 | 0.0015 |
| African Americans (n = 3.737) | 0.925 | 0.881 | 0.971 | 0.0015 | Reference | | | |
| Asians (n = 789) | 0.627 | 0.56 | 0.702 | <0.0001 | 0.681 | 0.604 | 0.768 | <0.0001 |
| Clinical T stage | | | | | | | | |
| cT2 | Reference | | | | | | | |
| cT3,4 | 1.133 | 1.003 | 1.28 | 0.0452 | | | | |
| Lymph node metastases | | | | | | | | |
| No | Reference | | | | | | | |
| Yes | 1.129 | 1.03 | 1.238 | 0.0099 | | | | |

HR: Hazard ration; 95% CI: 95% confidence interval.

metastases, and 10 months (95% CI 9.4–10.6) for liver metastases. In the bone metastases cohort, the survival probability at the 12[th] month was 72.2% (95% CI 76.7–77.7), 66.4% (95% CI 65.3–67.5) for the lung metastases cohort, 46.3% (95% CI 44.9–47.7) for the brain metastases cohort, and 46.2% (95% CI 45–47.4) for the liver metastases cohort (Fig 1 and S1 File). In Cox regression models, for patients with one site of visceral metastases, mPC patients with brain [Hazard ratio (HR) 1.692, 95% CI 1.623–1.765, p<0.0001] and liver metastases (HR 1.671, 95% CI 1.595–1.749, p<0.0001) experienced a poorer prognosis when compared to the lung metastases patients. There was no statistical difference seen in OS between the brain and liver metastases cohorts (Table 2).

## Impact of two visceral metastatic sites on outcomes

We subsequently evaluated the outcomes for mPC patients who had two visceral metastases. The median OS was 8 months (95%CI 6.8–9.2) for patients with brain + lung metastases, 6.6

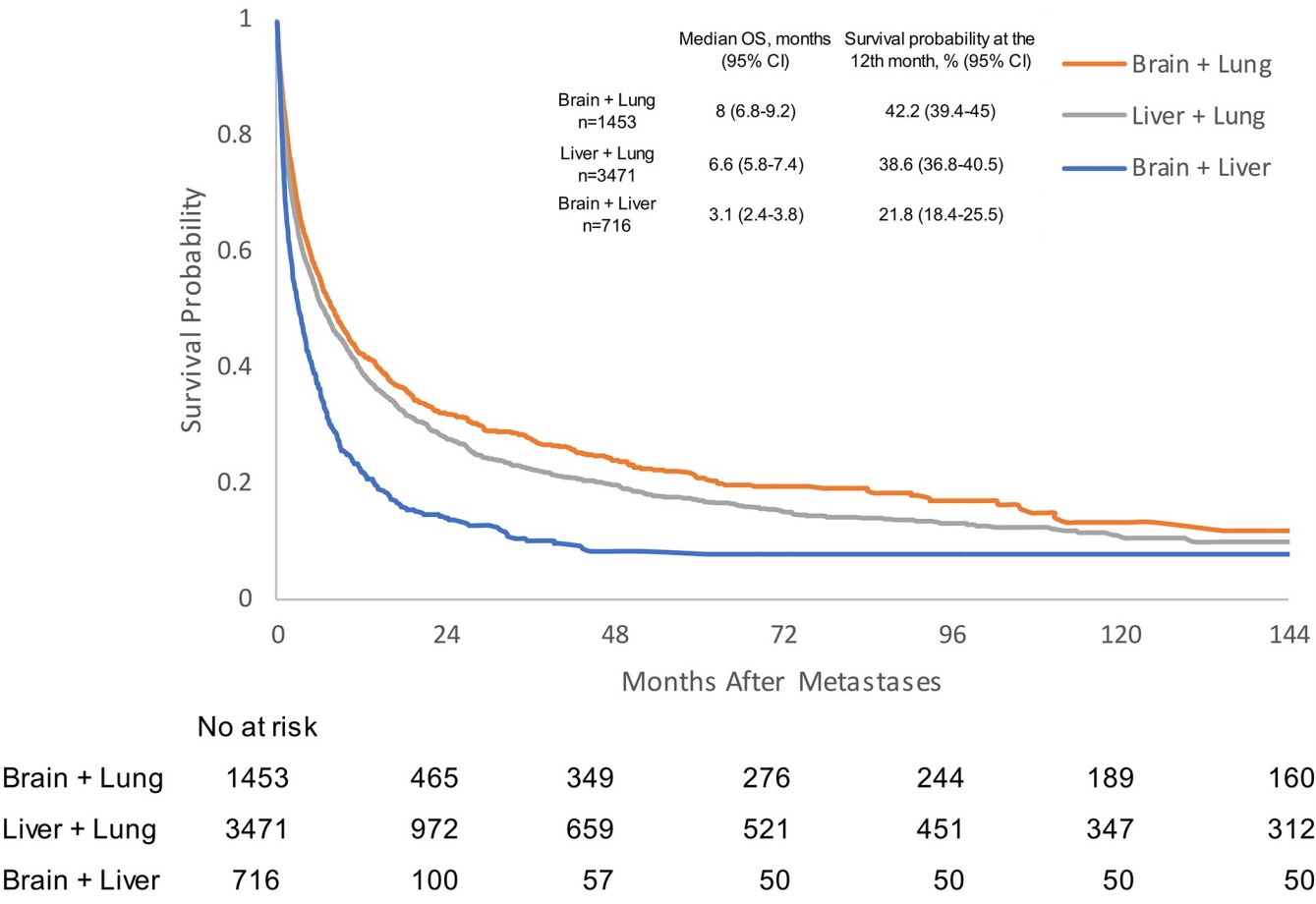

**Fig 2. Kaplan-Meier analysis of overall survival for prostate cancer patients with two visceral metastatic sites.** CI: confidence interval; OS: overall survival.

months (95%CI 5.8–7.4) for liver + lung metastases, and 3.1 months (95% CI 2.4–3.8) for liver + brain metastases (Fig 2 and S2 File). In Cox regression models, patients diagnosed with brain + lung metastases experienced better outcomes when compared to the liver + lung metastases (HR 1.138, 95% CI 1.054–1.23, p<0.0001) and liver + brain metastases patients (HR 1.707, 95% CI 1.529–1.906, p<0.0001). Moreover, liver + brain metastases (HR 1.488, 95% CI 1.349–1.64, p<0.0001) were associated with a poor OS when compared to liver + lung metastases (Table 2).

### Impact of concomitant non-visceral metastases on outcomes

In mPC patients with visceral metastases and concomitant bone metastases, median OS was 17 months (95% CI 14.5–19.1) for lung + bone metastases, 8.2 months (95% CI 7.2-.8) for brain + bone metastases, and 5.4 months (95% CI 4.8–6.2) for liver + bone metastases (Fig 3 and S3 File). Patients with liver + bone metastases experienced a worse OS when compared with brain + bone (HR 1.27, 95% CI 1.164–1.138, p<0.0001) and lung + bone metastases patients (HR 1.643, 95% CI 1.524–1.772, p<0.0001) (Table 2).

In mPC patients with visceral metastases and concomitant lymph node metastases, median OS was 42.2 months (95% CI 0.8–54.1) for lung + lymph node metastases, 13.5 months (95% CI 6.9–19.9) for brain + lymph node metastases, and 12.1 months (95% CI 9.9–13.7) for liver + lymph node metastases (Fig 4 and S4 File). Lung + lymph node metastases was shown to be

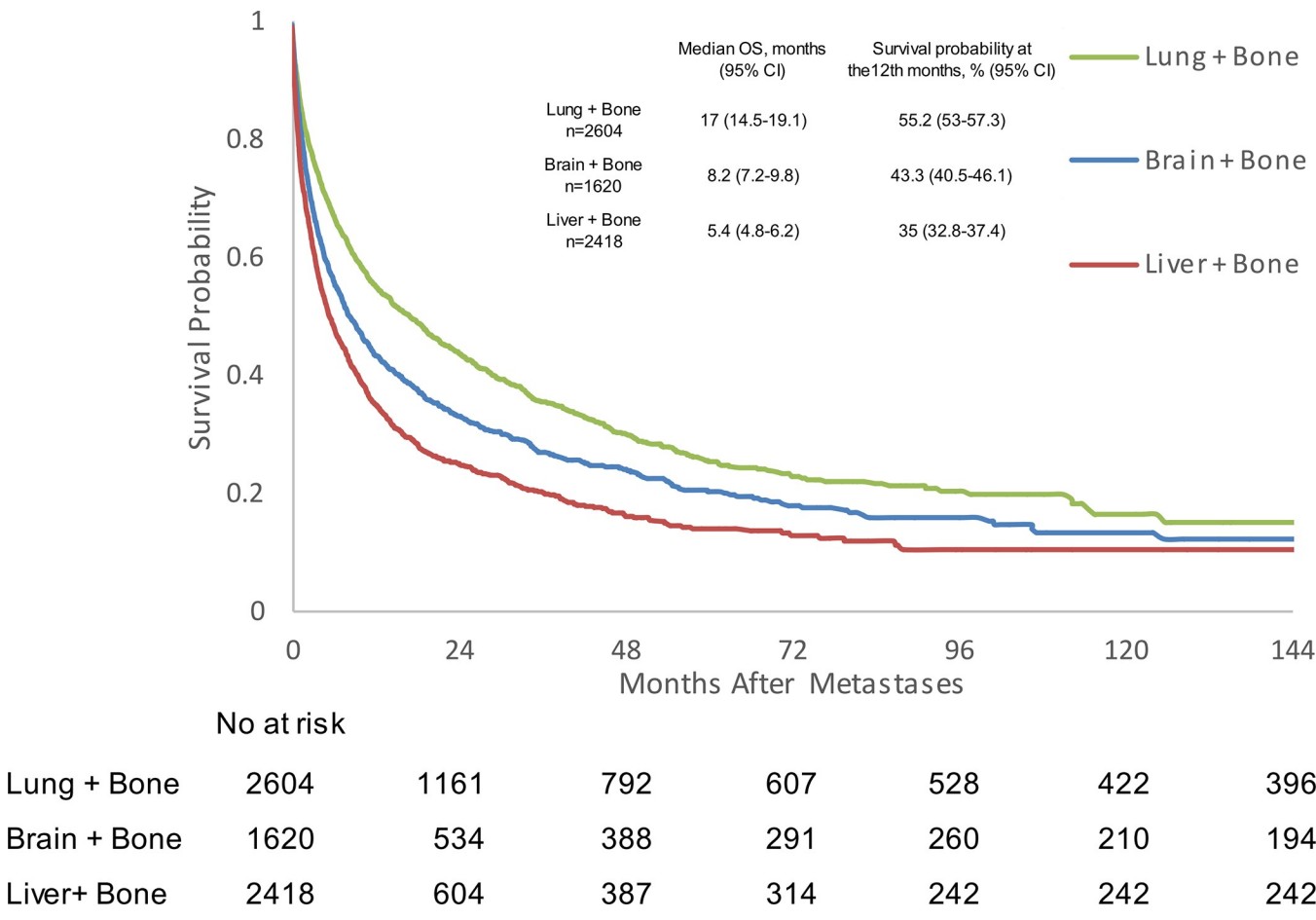

**Fig 3. Kaplan-Meier analysis of overall survival for prostate cancer patients with bone and visceral metastases.** CI: confidence interval; OS: overall survival.

a factor for better OS when compared with brain + lymph node (HR 1.63, 95% CI 1.361–1.952, p<0.0001) and liver + lymph nodes metastases (HR 1.706, 95% CI 1.513–1.922, p<0.0001). There was no significant difference seen in OS between brain + lymph node and liver + lymph node metastases (Table 2).

In mPC patients with visceral metastases and concomitant bone and lymph node metastases, median OS was 16 months (95% CI 13.6–18.1) for lung + bone + lymph node metastases, 8 months (95% CI 6.8–10.7) for brain + bone + lymph node metastases, and 5.6 months (95% CI 4.6–6.7) for liver + bone + lymph node metastases (Fig 5 and S5 File). Patients with liver + bone + lymph node metastases experienced a worse survival outcome when compared with both brain + bone + lymph node (HR 1.187, 95% CI 1.056–1.334, p = 0.004) and lung + bone + lymph node metastases (HR 1.641, 95% CI 1.499–1.796, p<0.0001) (Table 2).

## Impact of race on outcomes

We conducted survival analyses for mPC patients with visceral metastases among different races (Fig 6 and S6 File) and found that Caucasian patients had a poorer OS when compared with African American (HR 0.925, 95% CI 0.881–0.971, p<0.0001) and Asian patients (HR 0.627, 95% CI 0.56–0.702, p<0.0001). Asian patients were associated with an improved survival compared with African American patients (HR 0.681, 95% CI 0.604–0.768, p<0.0001) (Table 2).

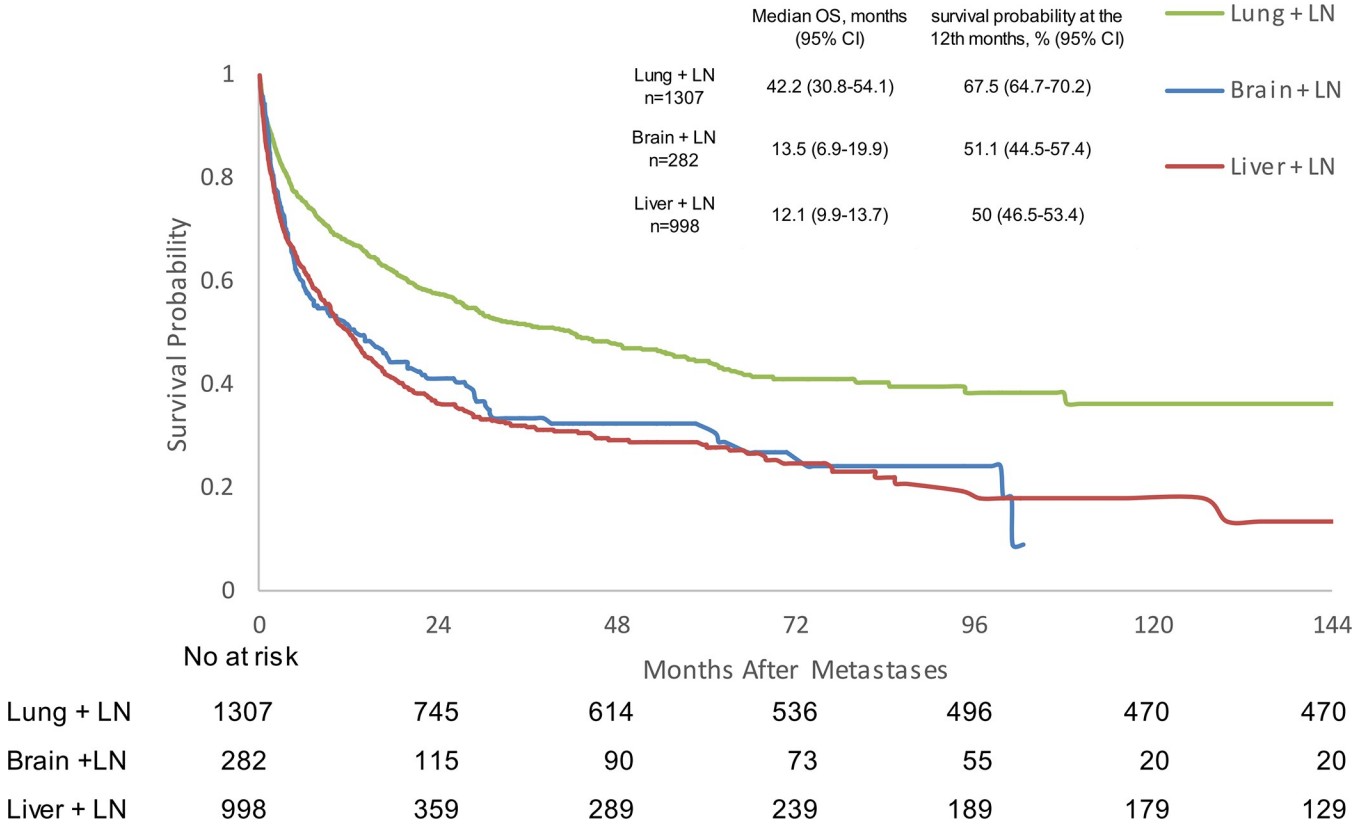

**Fig 4. Kaplan-Meier analysis of overall survival for prostate cancer patients with lymph node and visceral metastases.** CI: confidence interval; LN: lymph node; OS: overall survival.

## Discussion

In his study, we performed a retrospective analysis to compare the impact of specific visceral metastases sites on survival in patients with metastatic prostate cancer in a real-world setting. In patients having one visceral metastatic site with or without lymph node metastases, lung metastases were associated with improved outcomes when compared with brain or liver metastases. When combined with bone metastases or two visceral metastatic sites, liver metastasis was a factor for the worst OS when compared with lung or brain metastases. For OS among different races, Caucasian patients experienced poorer survival, while Asian patients experienced better outcomes.

In this study, we found that mPC patients with liver metastases had a median OS of 10 months. Liver metastases were a factor for worse outcomes than lung and non-visceral metastases. These findings are consistent with the prior studies, which had reported a median OS of 9–14 months for mPC patients with liver metastases, with these patients presenting as a group having a poor prognosis [8–13].

Additionally, we found that patients with lung metastases exhibited a worse OS than those with non-visceral metastases (bone or lymph node). The results are similar to the previous studies and further confirm the poorer outcomes which result from the presence of visceral metastases than those seen in non-visceral metastases [8–12].

Limited data regarding outcomes of prostate cancer with brain metastases have been reported, which may be attributed to the lower incidence of brain metastases. In 2018, Shou et al., in a population-based study using the Surveillance, Epidemiology, and End Results

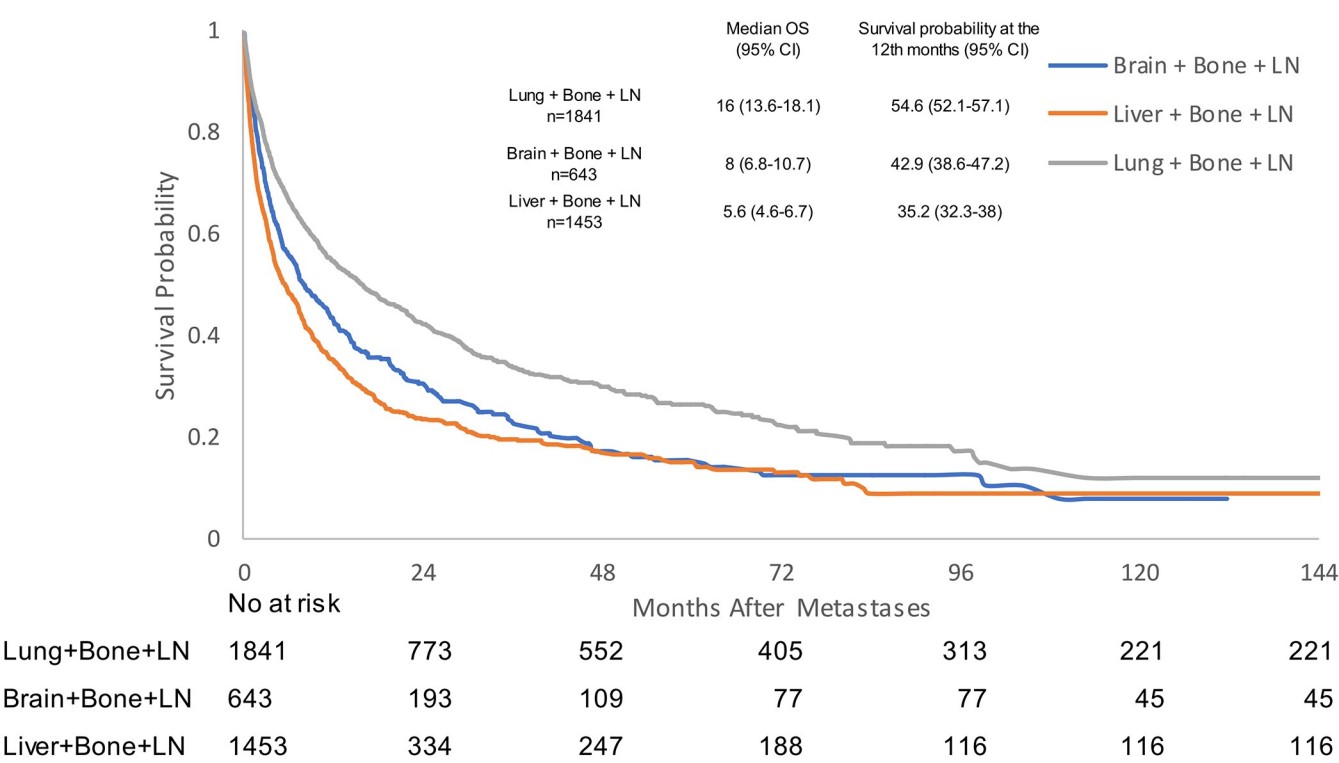

**Fig 5. Kaplan-Meier analysis of overall survival for prostate cancer patients with bone, lymph node, and visceral metastases.** CI: confidence interval; LN: lymph node; OS: overall survival.

(SEER) database, demonstrated that patients with brain metastases may exhibit better outcomes than those with liver metastases [11]. In 2023, Tappero et al., using the SEER database, reported a poor prognosis for liver and brain metastases when compared with lung metastases, with no difference being seen in survival between liver and brain metastases [13]. In this study, we found that liver and brain metastases were similar in OS for mPC patients with only one visceral metastases site. However, for mPC patients with two visceral metastatic sites or one visceral metastatic site with concomitant bone metastases, brain metastases may be associated with improved survival when compared with liver metastases. These findings may provide clinicians with better risk stratification and evaluation for these patients.

In this study, we found that Asian patients with visceral metastases experienced survival advantages more so than Caucasian and African American patients. This finding is consistent with the available prior literature, which shows Asian patients with metastatic prostate cancer having better OS [14–17]. The differences in genomics, lifestyle and/or treatment responsiveness may have contributed to these results. The survival outcomes seen between African American and Caucasian patients with mPC are controversial. Several studies have reported improved survival rates for African American patients when compared to Caucasian patients when receiving therapies [14, 18–22]. Our study also revealed that African American patients with visceral metastatic prostate cancer exhibited better OS than Caucasian patients. This may be explained by the increased immune response in African American patients during treatments when compared to the Caucasian patients [14].

There were limitations in our study. First, the retrospective design of the present study may have led to selection bias. Second, we could not assess the information regarding disease burden, such as the number or size of a specific visceral metastasis. Third, the detailed data, such

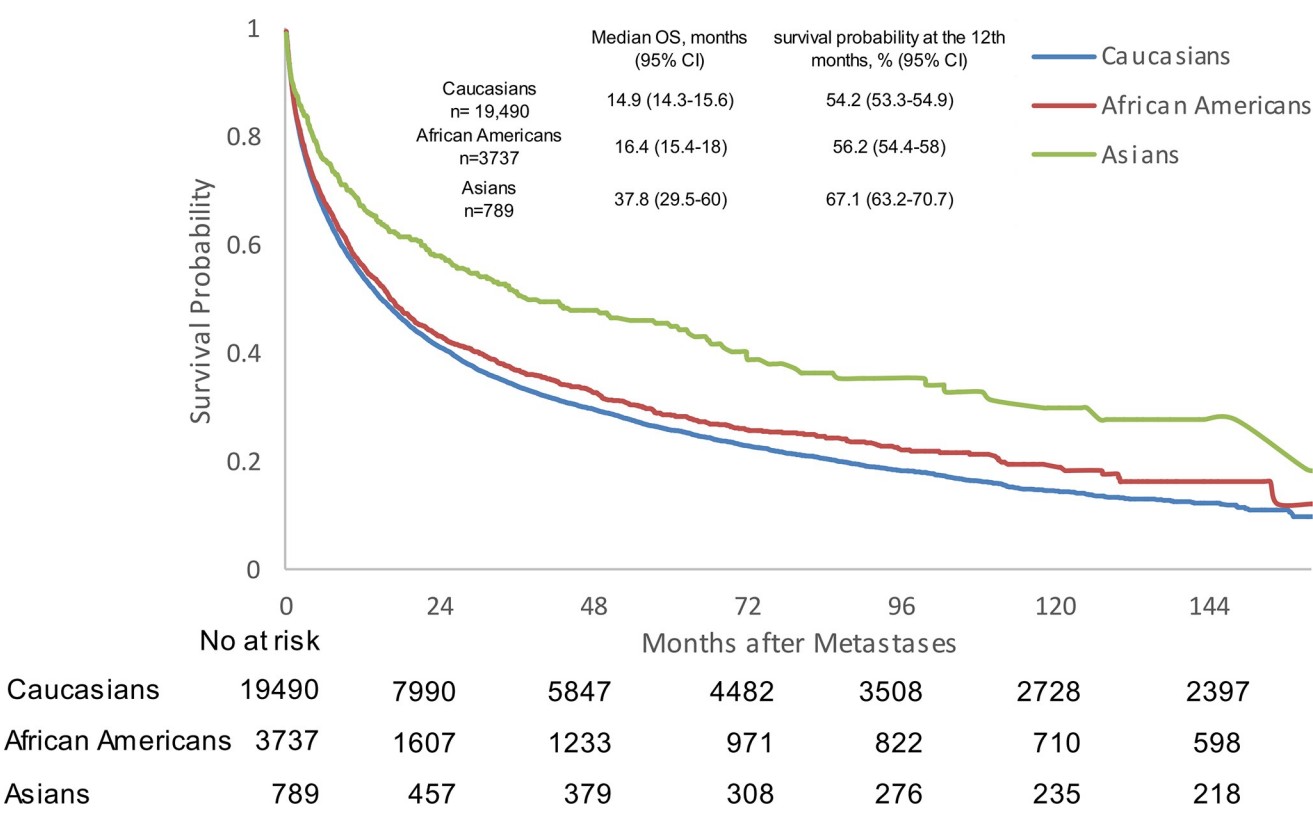

**Fig 6. Kaplan-Meier analysis of overall survival among different races for patients with visceral metastatic prostate cancer.** CI: confidence interval; OS: overall survival.

as castration sensitive or resistant prostate cancer, BRCA gene mutation status, or family history of prostate cancer, were either not available or inadequate for analysis from the database. Fourth, our research lacked the information on methodology for diagnosis and follow-up, such as computed tomography, magnetic resonance imaging, bone scan or positron emission tomography, and different image tools may affect the diagnostic rate of metastases. Despite these limitations, our analysis included a large population diagnosed with visceral metastatic prostate cancer, and the results may offer clinicians useful information toward predicting a patient's prognosis or for the better design of trials for these patients.

## Conclusion

For prostate cancer patients with visceral metastases, patients with lung metastases experience survival benefits when compared to those with either liver or brain metastases. When combining two visceral metastatic sites or concomitant bone metastases, liver metastases were associated with worse outcomes. Asian patients experienced better OS than both Caucasian and African American patients diagnosed with visceral metastatic prostate cancer.

## Supporting information

**S1 File. Values for Kaplan-Meier survival analysis in prostate cancer patients with bone or one visceral metastatic site.**
(XLSX)

**S2 File. Values for Kaplan-Meier survival analysis in prostate cancer patients with two visceral metastatic sites.**
(XLSX)

**S3 File. Values for Kaplan-Meier survival analysis in prostate cancer patients with bone and visceral metastases.**
(XLSX)

**S4 File. Values for Kaplan-Meier survival analysis in prostate cancer patients with lymph node and visceral metastases.**
(XLSX)

**S5 File. Values for Kaplan-Meier survival analysis in prostate cancer patients with bone, lymph node, and visceral metastases.**
(XLSX)

**S6 File. Values for Kaplan-Meier survival analysis among different races in patients with visceral metastatic prostate cancer.**
(XLSX)

## Acknowledgments

The data used in this research was from the TriNetX network.

## Author Contributions

**Conceptualization:** Gu-Shun Lai, Jason Chia-Hsien Cheng, Shian-Shiang Wang.

**Data curation:** Gu-Shun Lai.

**Formal analysis:** Gu-Shun Lai, Shian-Shiang Wang.

**Investigation:** Gu-Shun Lai.

**Methodology:** Gu-Shun Lai, Jian-Ri Li, Shian-Shiang Wang.

**Writing – original draft:** Gu-Shun Lai, Shian-Shiang Wang.

**Writing – review & editing:** Gu-Shun Lai, Chuan-Shu Chen, Jason Chia-Hsien Cheng, Jian-Ri Li, Cheng-Kuang Yang, Chia-Yen Lin, Sheng-Chun Hung, Kun-Yuan Chiu, Shian-Shiang Wang.

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
