## [Decision Letter · Decision Letter 0]

29 Jul 2024

PONE-D-24-15232Impact of different visceral metastatic sites on survival in metastatic prostate cancer patientsPLOS ONE

Dear Dr. Lai,

Thank you for submitting your manuscript to PLOS ONE. After careful consideration, we feel that it has merit but does not fully meet PLOS ONE’s publication criteria as it currently stands. Therefore, we invite you to submit a revised version of the manuscript that addresses the points raised during the review process. Please submit your revised manuscript by Sep 12 2024 11:59PM. If you will need more time than this to complete your revisions, please reply to this message or contact the journal office at plosone@plos.org. Please include the following items when submitting your revised manuscript:A rebuttal letter that responds to each point raised by the academic editor and reviewer(s). You should upload this letter as a separate file labeled 'Response to Reviewers'.A marked-up copy of your manuscript that highlights changes made to the original version. You should upload this as a separate file labeled 'Revised Manuscript with Track Changes'.An unmarked version of your revised paper without tracked changes. You should upload this as a separate file labeled 'Manuscript'.If applicable, we recommend that you deposit your laboratory protocols in protocols.io to enhance the reproducibility of your results. Protocols.io assigns your protocol its own identifier (DOI) so that it can be cited independently in the future. For instructions see: https://journals.plos.org/plosone/s/submission-guidelines#loc-laboratory-protocols. Additionally, PLOS ONE offers an option for publishing peer-reviewed Lab Protocol articles, which describe protocols hosted on protocols.io. Read more information on sharing protocols at https://plos.org/protocols?utm_medium=editorial-email&utm_source=authorletters&utm_campaign=protocols.

We look forward to receiving your revised manuscript.

Kind regards,

Matteo Bauckneht

Academic Editor

PLOS ONE

Reviewers' comments:

Reviewer's Responses to Questions

**Comments to the Author**

1. Is the manuscript technically sound, and do the data support the conclusions?

Reviewer #1: Yes

Reviewer #2: Yes

2. Has the statistical analysis been performed appropriately and rigorously? 

Reviewer #1: Yes

Reviewer #2: Yes

3. Have the authors made all data underlying the findings in their manuscript fully available?

Reviewer #1: Yes

Reviewer #2: Yes

4. Is the manuscript presented in an intelligible fashion and written in standard English?

Reviewer #1: Yes

Reviewer #2: Yes

5. Review Comments to the Author

Reviewer #1: The authors performed a very interesting analysis based on the work of Tappero et al. using the SEER databases. They successfully demonstrated that different metastatic sites serve as predictors of varying survival rates in patients with metastatic prostate cancer (PCa). The authors can boast a large population, solid methodology, and results that align with the previous work of Tappero, despite being based on a different population and database. The paper is well-written and easily readable. The tables and figures are well-crafted, presenting fundamental information without redundancy. I particularly appreciated the summary of median survival within the Kaplan-Meier analysis.

However, I have a few suggestions for improvement:

1. In the analysis of race/ethnicity, I recommend using the terms "African Americans" and "Caucasians" instead of "Black" and "White," respectively. Additionally, please note that "Black" was incorrectly reported as "Blake" in the summary.

2. I suggest adding the lack of information on the BRCA status of patients as a limitation.

3. Additionally, the absence of data on family history of PCa should be mentioned among the limitations.

4. The limitations should also include the absence of details regarding the modality of diagnosis and follow-up (i.e., CT scan, bone scintigraphy, PET-CT scan).

A major revision is required to address these points.

Reviewer #2: Authors' work is interesting. I read with pleasure the full manuscript, written in good English. It is noteworthy to implement the limitation section. Indeed, data regarding the primary diagnosis methodology, such as CT, PET should be added. With minor revision, the paper will be suitable for publication

6. PLOS authors have the option to publish the peer review history of their article (what does this mean?). If published, this will include your full peer review and any attached files.

Reviewer #1: **Yes: **Nicola Longo

Reviewer #2: **Yes: **SIMONE MORRA

---

## [Author Response · Author response to Decision Letter 0]

7 Aug 2024

Dear Reviewers:

 I am deeply grateful for the time and effort you dedicated to reviewing my article. Your feedback and thoughtful questions have been incredibly valuable in refining my work. I appreciate your constructive suggestions and am committed to addressing each point you raised as follows.

Reviewer #1: 

1. In the analysis of race/ethnicity, I recommend using the terms "African Americans" and "Caucasians" instead of "Black" and "White," respectively. Additionally, please note that "Black" was incorrectly reported as "Blake" in the summary.

The “Blake” and “White” have changed to "African Americans" and "Caucasians", respectively, in the revised manuscript. Thank you for your recommendation.

2. I suggest adding the lack of information on the BRCA status of patients as a limitation.

The lack of data on BRCA mutation status of patients has been added in the limitation section. Thank you for your recommendation.

3. Additionally, the absence of data on family history of PCa should be mentioned among the limitations.

The absence of information on family history of prostate cancer has been added in the limitation section. Thank you for your recommendation.

4. The limitations should also include the absence of details regarding the modality of diagnosis and follow-up (i.e., CT scan, bone scintigraphy, PET-CT scan).

The lack of data regarding the methodology for diagnosis and follow-up has been added in the limitation section. Thank you for your recommendation.

Reviewer #2: 

The lack of data regarding the methodology for diagnosis and follow-up has been added in the limitation section. Thank you for your recommendation.

---

## [Decision Letter · Decision Letter 1]

22 Aug 2024

Impact of different visceral metastatic sites on survival in metastatic prostate cancer patients

PONE-D-24-15232R1

Dear Dr. Lai,

We’re pleased to inform you that your manuscript has been judged scientifically suitable for publication and will be formally accepted for publication once it meets all outstanding technical requirements.

Kind regards,

Matteo Bauckneht

Academic Editor

PLOS ONE

Additional Editor Comments (optional):

Reviewers' comments:

Reviewer's Responses to Questions

**Comments to the Author**

1. If the authors have adequately addressed your comments raised in a previous round of review and you feel that this manuscript is now acceptable for publication, you may indicate that here to bypass the “Comments to the Author” section, enter your conflict of interest statement in the “Confidential to Editor” section, and submit your "Accept" recommendation.

Reviewer #2: All comments have been addressed

2. Is the manuscript technically sound, and do the data support the conclusions?

Reviewer #2: Yes

3. Has the statistical analysis been performed appropriately and rigorously? 

Reviewer #2: Yes

4. Have the authors made all data underlying the findings in their manuscript fully available?

Reviewer #2: Yes

5. Is the manuscript presented in an intelligible fashion and written in standard English?

Reviewer #2: Yes

6. Review Comments to the Author

Reviewer #2: The authors exhaustively replied to all the concerns raised. Therefore, the manuscript is suitable in its current form

7. PLOS authors have the option to publish the peer review history of their article (what does this mean?). If published, this will include your full peer review and any attached files.

Reviewer #2: **Yes: **SIMONE MORRA

---

## [Editor Report · Acceptance letter]

29 Aug 2024

PONE-D-24-15232R1 

PLOS ONE

Dear Dr. Lai, 

I'm pleased to inform you that your manuscript has been deemed suitable for publication in PLOS ONE. Congratulations! Your manuscript is now being handed over to our production team.

Kind regards, 

on behalf of

Dr. Matteo Bauckneht 

Academic Editor

PLOS ONE